# Interdisciplinary Collaboration on Real World Data to Close the Knowledge Gap: A Reflection on “De Sutter et al. Predicting Volume of Distribution in Neonates: Performance of Physiologically Based Pharmacokinetic Modelling”

**DOI:** 10.3390/pharmaceutics16010128

**Published:** 2024-01-19

**Authors:** Karel Allegaert, Anne Smits, Pieter Annaert

**Affiliations:** 1Clinical Pharmacology and Pharmacotherapy, Department of Pharmaceutical and Pharmacological Sciences, KU Leuven, 3000 Leuven, Belgium; 2Department of Development and Regeneration, KU Leuven, 3000 Leuven, Belgium; anne.smits@uzleuven.be; 3Department of Hospital Pharmacy, Erasmus University Medical Center, 3000 CA Rotterdam, The Netherlands; 4Neonatal Intensive Care Unit, University Hospitals Leuven, 3000 Leuven, Belgium; 5Drug Delivery and Disposition, Department of Pharmaceutical and Pharmacological Sciences, KU Leuven, 3000 Leuven, Belgium; pieter.annaert@kuleuven.be

**Keywords:** physiologically based pharmacokinetics (PBPK), pharmacokinetics, neonates, volume of distribution, allometric scaling, developmental pharmacology, real world data

## Abstract

This commentary further reflects on the paper of De Sutter et al. on predicting volume of distribution in neonates, and the performance of physiologically based pharmacokinetic models We hereby stressed the add on value to collaborate on real world data to further close this knowledge gap. We illustrated this by weight distribution characteristics in breastfed (physiology) and in asphyxiated (pathophysiology), with additional reflection on their kidney and liver function.

With great interest, we have read the analysis reported by De Sutter et al. on the performance of physiologically based pharmacokinetic (PBPK) modeling to predict the distribution volume (Vd) in (pre)term neonates [1]. We hereby value the original approach taken to focus on Vd characteristics as this PK parameter is understudied, while important to estimate elimination half-life or to support loading dose practices. For the majority of drugs, newborns combine a proportionally (to body weight) higher Vd, with a clinically relevant lower clearance (CL) [2]. The key differences in PK parameters (Vd, CL) in the newborn and the adult are illustrated in Figure 1.

This combination (higher Vd and lower clearance) further prolongs the elimination half-life of many drugs in neonates and makes loading dose practices with proportionally higher (mg/kg) doses more relevant if a given threshold concentration should be reached in a timely fashion (e.g., analgesics, sedatives, anti-epileptics, and antibiotics). “Underloading” will hereby result in poorer effects (therapy failure), and “overloading” will result in overexposure, with subsequent delayed elimination (clearance lower and elimination half-life prolonged) and potential side effects or toxicity. Therefore, accurate estimates of Vd in neonates are of specific relevance in this population [2].

In our reading, De Sutter et al. illustrated that simple isometric scaling to adult values (/kg) resulted in relevant and systematic underestimation (average fold error 0.61) of the “true/observed” distribution volume. Because of the impact of loading doses and the elimination half-life, this is of relevance for both clinical trials and neonatal pharmacotherapy. Failure to use simple isometric scaling is indeed not unexpected as body composition differs significantly [1].

This underprediction improved when PBPK-based prediction methods (Poulin and Theil, with Berezhovsky correction, or Rodgers and Rowland) (average fold error of 0.82–0.83) were applied. Interestingly, the Rodgers and Rowland correction outperformed the Poulin and Theil method when considering the proportion of drugs (71 versus 54%) within the twofold range, while—when inaccurate—the Rodgers and Rowland related errors were larger than those of the Poulin and Theil approach. This caused the authors to conclude that their results highlighted both the applicability and limitations of PBPK methods to predict Vd in neonates in the absence of clinical data [1].

It is especially this last sentence that triggered us to write this letter as the absence of clinical data in their paper focused on the absence of concentration–time profiles as PBPK models are commonly used during drug development plans before such profiles become available. However, we claim that these limitations at least in part relate to knowledge gaps, giving rise to various levels of PBPK model uncertainty. Indeed, there are still many additional unexplored opportunities to inform and improve PBPK modeling efforts in (pre)term neonates with characteristics of the neonatal populations and (patho)physiological data.

This is the area of interdisciplinary collaboration that we aim to highlight as clinicians are not always aware of the relevance of real-world data. Once converted to mathematical functions, such data can be very instrumental to further improve PBPK. It is a fact that PBPK model development papers commonly mention that the developers of PBPK models indeed lacked longitudinal system parameters or data on physiological parameters to “feed” the system [3]. To further illustrate this, Abduljalil developed a preterm PBPK model mainly driven by fetal physiology and clinical data, not likely similar to the “real” postnatal patterns [3].

In the de Sutter paper, the median body weight was either inputted from the source data (the published population PK papers) or otherwise arbitrarily set at 1.19 kg for a preterm, 3.23 kg for a term, or 2.44 kg for a mixed population [1]. To improve this arbitrary setting, we as a clinical research community have to provide relevant information on such characteristics (like median, as well as range in weight) of treated (sub)populations to the public domain. Kimko et al. already illustrated that the use of non-disease-specific growth or weight charts for pediatric populations has major limitations [4]. This is because the weight distribution of the subjects enrolled in actual clinical trials depends on the type of the disease of interest, commonly different from non-disease-specific growth charts [4].

To illustrate its feasibility and relevance for a very specific subpopulation of neonates, we provide the birth weight distribution characteristics of a recently published cohort of neonates that underwent therapeutic hypothermia after moderate to severe asphyxia [5]. Compared with a reference cohort, the median and interquartile birth weights (3.35, 2.93–3.74 kg) were very similar to those of a reference cohort of (near-) term neonates [6]. However, the portion of neonates < 2.5 kg or >4 kg was 8.8% (instead of 6.3%) and 15.2% (instead of 8.3%), respectively. When calculated based on being small for gestational age (SGA, birth weight < 10th centile) or large for gestational age (LGA, birth weight > 90th centile), these were 18.4 and 14.8%, respectively, instead of the expected 10% [6]. Such real-world data (RWD) are relevant to calculate the expected Vd (absolute value, L) and its variability and may assist to develop drug dosing regimens specific to this population and its variability. Similar efforts can be considered for other PK relevant variables, like kidney function, cardiac output and regional blood flow, or blood/plasma composition [7].

In addition to weight, specific (patho)physiological conditions should be considered as diseases related covariates as they may have impacts on Vd characteristics. This has been highlighted by de Sutter et al. as they excluded studies during extracorporeal membrane oxygenation or cardiac bypass based on the fact that these interventions affected the Vd [1]. Along similar lines, the Vd of aminoglycosides in neonates with sepsis is significantly higher when compared with that in non-septic neonates. Finally, raised indirect hyperbilirubinemia is a common medical condition in neonates. This changes Vd by affecting drug albumin binding capacity, as illustrated for, e.g., ibuprofen [8,9].

In addition to disease-related covariates, we also would like to refer to an ongoing effort to compare and map the physiology and body composition of breastfed to formula-fed infants within the IMI Conception consortium, hereby using systematic search strategies [10]. There are longitudinal differences in, e.g., weight, length, and body composition (lean body mass, fat mass) between both populations, so that the type of feeding will result in a given trait [10]. These differences may be useful to explore different scenarios and become most relevant when PBPK models are applied to quantify lactation-related drug exposure associated with maternal pharmacotherapy [10].

PBPK modeling and simulation provides a structured approach and holds the promise to support drug development in neonates. This approach has already supported regulatory filings and is commonly used during drug development in this population. We anticipate that it will be a key contributor to regulatory drug approvals for use in neonates [11]. The findings of de Sutter et al. hereby highlight both the applicability and limitations of PBPK methods to predict Vd in neonates [1]. We claim that these limitations are at least in part related to knowledge gaps on real-world data, converted to mathematical functions. To improve this, we need cross talk and interdisciplinary collaboration between clinicians, clinical researchers, and modelers to generate and integrate knowledge (PK datasets, system knowledge, and maturational (patho)physiology) to further refine these PBPK models and finally improve their predictive performance.

## Figures and Tables

**Figure 1 pharmaceutics-16-00128-f001:**
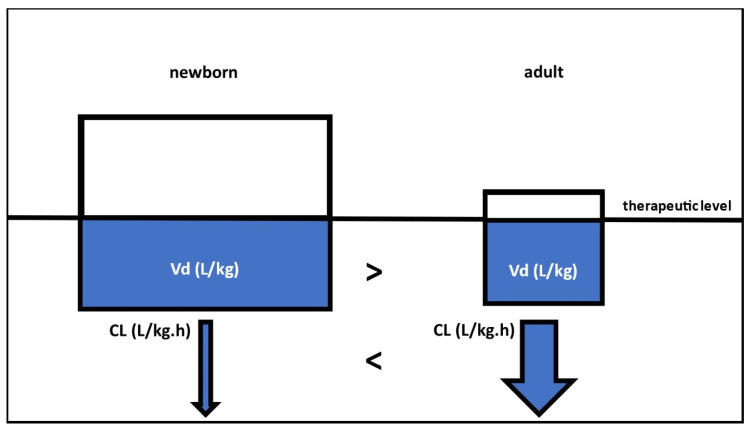
Schematic comparison in key pharmacokinetic parameters (distribution volume, Vd; and clearance, CL) between a newborn (**left**) and a reference adult (**right**). To reach a similar therapeutic level, a proportionally higher (mg/kg) loading dose is needed in neonates compared with adults.

## Data Availability

Not applicable.

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
