# Peer review of "Interdisciplinary Collaboration on Real World Data to Close the Knowledge Gap: A Reflection on “De Sutter et al. Predicting Volume of Distribution in Neonates: Performance of Physiologically Based Pharmacokinetic Modelling”"

_pharmaceutics, 2024, doi:10.3390/pharmaceutics16010128_

Round 1

Reviewer 1 Report

Comments and Suggestions for Authors

I found this letter which comments on a recently published paper "De Sutter et al. Predicting Volume of Distribution in Neonates: Performance of Physiologically Based Pharmacokinetic Modelling" was well presented, and easy to understand. The authors pointed out the importance of incorporating clinical "knowledge" e.g., neonatal body weights owing to diseases or other covariants may already be available. Cross-dialogues are indeed necessary for such multidisciplinary work as PBPK.

Author Response

We thank this reviewer for the very positive and supportive assessment of our letter/commentary. As there are no specific comments for revision, we have not changed anything in the letter. 

Reviewer 2 Report

Comments and Suggestions for Authors

Comments for the authors

The authors present a well written commentary related to a recent publication of DeSutter related to the prediction of the volume of distribution in neonates.  This is an important and understudied population and the authors of the commentary highlight the continuing challenges related to persistent prediction biases with current approaches reflected in the DeSutter article.  The authors highlight that these biases may result in improper dosing in both the loading dose and maintenance dose scenarios thus putting neonatal patients at risk.  Specific comments are enumerated below.

1.     The authors suggest the role of interdisciplinary cooperation as the key means of addressing continuing issues with prediction bias in approaches to understanding the neonatal volume of distribution (and the follow on effects on dosing resulting in improper drug exposures).  However, this suggestion is mainly highlighted in the title and the concluding paragraph of the commentary.  It would be helpful for the reader if this was more explicitly addressed – i.e., in a very specific way how do each of the disciplines stand to contribute to the gaps in knowledge so eloquently expressed in this manuscript.  How do the authors propose that the different disciplines should be convened / encouraged / facilitated in working together to address specific gaps.

2.     Line 25, the authors state “proportionally higher” – perhaps further clarify – i.e., “proportionally higher relative to body weight…”

3.     Line 34 – the authors state “reached timely” in relation to achieving target concentrations.  Perhaps “reached in a timely fashion” would be more appropriate.

4.     Lines 40-47 – the authors highlight the prediction ratios of less than 1 for volume of distribution.  A recapitulation on the impacts with respect to both loading doses and maintenance doses (based on calculation of half-life etc.) may be helpful for the reader – explicitly quantifying the departure/deviation that would be expected in the target concentrations based on the use of these biased predictions of volume could be one strategy for further refining the message of this paragraph (especially for a clinician target audience).

5.     Line 61 – consider changing “for sure not…” to “not likely to be…”

6.     Lines 88-92 – the discussion of indirect bilirubinemia could be clarified – the extra statement of “very common condition in neonates” detracts from the effect on albumin binding capacity point that is being made later in the sentence – perhaps indirect bilirubinemia changes Vd by affecting drug albumin binding capacity.

7.     Lines 101- the final paragraph -the initial statement “track record up to regulatory acceptance …” is somewhat unclear.  Perhaps highlighting the PBPK modeling first – i.e., “PBPK modelling and simulation provides a structured approach to understanding drug disposition and holds the promise to support drug development in neonates.  This approach has supported regulatory filings … (include evidence here for applications where this has specifically been utilized – scaling in non-neonate pediatrics, ddi assessment reducing studies needed etc.) and we anticipate that it will be a key contributor to regulatory drug approvals for use in neonates.”

Comments on the Quality of English Language

Adequate

Author Response

  The authors suggest the role of interdisciplinary cooperation as the key means of addressing continuing issues with prediction bias in approaches to understanding the neonatal volume of distribution (and the follow on effects on dosing resulting in improper drug exposures).  However, this suggestion is mainly highlighted in the title and the concluding paragraph of the commentary.  It would be helpful for the reader if this was more explicitly addressed – i.e., in a very specific way how do each of the disciplines stand to contribute to the gaps in knowledge so eloquently expressed in this manuscript.  How do the authors propose that the different disciplines should be convened / encouraged / facilitated in working together to address specific gaps.

We understood that there is not much ‘flexibility’ on the title, as this is a letter in response to a previously published paper. However, we have further extended the ‘mid’ section of the letter, with focus on how clinicians can contribute to PBPK by generating RWD, converted to mathematical functions.

  1. Line 25, the authors state “proportionally higher” – perhaps further clarify – i.e., “proportionally higher relative to body weight…”

adapted

  1. Line 34 – the authors state “reached timely” in relation to achieving target concentrations.Perhaps “reached in a timely fashion” would be more appropriate.

adapted

  1. Lines 40-47 – the authors highlight the prediction ratios of less than 1 for volume of distribution.A recapitulation on the impacts with respect to both loading doses and maintenance doses (based on calculation of half-life etc.) may be helpful for the reader – explicitly quantifying the departure/deviation that would be expected in the target concentrations based on the use of these biased predictions of volume could be one strategy for further refining the message of this paragraph (especially for a clinician target audience).

adapted

  1. Line 61 – consider changing “for sure not…” to “not likely to be…”

adapted

  1. Lines 88-92 – the discussion of indirect bilirubinemia could be clarified – the extra statement of “very common condition in neonates” detracts from the effect on albumin binding capacity point that is being made later in the sentence – perhaps indirect bilirubinemia changes Vd by affecting drug albumin binding capacity.

Rephrased.

  1. Lines 101- the final paragraph -the initial statement “track record up to regulatory acceptance …” is somewhat unclear.Perhaps highlighting the PBPK modeling first – i.e., “PBPK modelling and simulation provides a structured approach to understanding drug disposition and holds the promise to support drug development in neonates.  This approach has supported regulatory filings … (include evidence here for applications where this has specifically been utilized – scaling in non-neonate pediatrics, ddi assessment reducing studies needed etc.) and we anticipate that it will be a key contributor to regulatory drug approvals for use in neonates.”

Rephrased.